# Leukocyte Count in Solid Organ Transplant Recipients After SARS-CoV-2 mRNA Vaccination and Infection

**DOI:** 10.3390/vaccines13020103

**Published:** 2025-01-22

**Authors:** Marita Kern, Sebastian Rask Hamm, Christian Ross Pedersen, Dina Leth Møller, Josefine Amalie Loft, Rasmus Bo Hasselbalch, Line Dam Heftdal, Mia Marie Pries-Heje, Michael Perch, Søren Schwartz Sørensen, Allan Rasmussen, Peter Garred, Kasper Karmark Iversen, Henning Bundgaard, Caroline A. Sabin, Susanne Dam Nielsen

**Affiliations:** 1Viro-Immunology Research Unit, Department of Infectious Diseases, Rigshospitalet, Copenhagen University Hospital, 2100 Copenhagen, Denmarksebastian.rask.hamm.02@regionh.dk (S.R.H.);; 2Department of Surgical Gastroenterology and Transplantation, Rigshospitalet, Copenhagen University Hospital, 2100 Copenhagen, Denmark; 3Department of Cardiology, Herlev and Gentofte Hospital, Copenhagen University Hospital, 2730 Herlev, Denmark; 4Department of Emergency Medicine, Herlev and Gentofte Hospital, Copenhagen University Hospital, 2730 Herlev, Denmark; 5Department of Cardiology, Rigshospitalet, Copenhagen University Hospital, 2100 Copenhagen, Denmark; 6Department of Clinical Medicine, Faculty of Health and Medical Sciences, University of Copenhagen, 2100 Copenhagen, Denmark; 7Department of Nephrology, Rigshospitalet, Copenhagen University Hospital, 2100 Copenhagen, Denmark; 8Laboratory of Molecular medicine, Department of Clinical Immunology, Rigshospitalet, Copenhagen University Hospital, 2100 Copenhagen, Denmark; 9Centre for Clinical Research, Epidemiology, Modelling and Evaluation, Institute for Global Health, UCL, Royal Free Campus, Rowland Hill St., London NW3 2PF, UK

**Keywords:** SARS-CoV-2 mRNA vaccination, SARS-CoV-2 infection, solid organ transplant recipient, leukocyte count, leukopenia

## Abstract

Background: Solid organ transplant (SOT) recipients are at risk of severe COVID-19. Vaccination is an important preventive measure but may have side effects, including decreased leukocyte counts. We aimed to describe the prevalence and relative incidence of decreased leukocyte counts and changes in leukocyte counts before and after SARS-CoV-2 mRNA vaccination and SARS-CoV-2 infection in SOT recipients. Methods: Changes in leukocyte counts from before to after each vaccine dose were investigated using linear mixed models. We determined the prevalence of decreased leukocyte counts before and after each vaccine dose and before and after SARS-CoV-2 infection. Self-controlled case series analysis was used to investigate whether the period after either vaccination or infection was associated with risk of decreased leukocyte count. Results: We included 228 adult kidney, lung, and liver transplant recipients. Prior to the first vaccine dose, the mean leukocyte count was 7.3 × 10^9^ cells/L (95% CI 6.9–7.6). Both the leukocyte counts, and the prevalence of decreased leukocyte counts remained unchanged from before to after vaccination regardless of the number of vaccine doses provided. There was no association between vaccination and decreased leukocyte counts (incidence rate ratio (IRR): 0.6; 95% CI: 0.2–2.1; *p* = 0.461). In contrast, SARS-CoV-2 infection was associated with increased risk of a decreased leukocyte count (IRR: 7.1; 95% CI: 2.8–18.1; *p* < 0.001). Conclusions: SARS-CoV-2 mRNA vaccination was not associated with risk of decreased leukocyte count and did not affect the prevalence of decreased leukocyte counts in SOT recipients. In contrast, SARS-CoV-2 infection was associated with a higher risk of a decreased leukocyte count.

## 1. Introduction

Solid organ transplant (SOT) recipients remain at increased risk of severe COVID-19 [1,2]. Vaccination is an effective and important strategy to mitigate this risk [3,4,5]. SARS-CoV-2 mRNA vaccines have been tested in SOT populations and shown safety profiles similar to those in the background population [3,6,7,8,9,10]. Furthermore, vaccination with BNT162b2 has also been reported to be safe in immunocompromised populations other than SOT recipients [10,11,12,13]. However, some studies in healthy individuals have reported adverse hematological events following vaccination. Most of these studies are case series of thrombocytopenia, but one case–control study reported an increased risk of leukopenia after the second dose of the BNT162b2 vaccine [14,15,16,17]. Additionally, two randomized controlled trials of the BNT162b1 vaccine found decreased lymphocyte counts in up to 50% of participants after the first dose [18,19]. Furthermore, a decrease in leukocyte count after influenza vaccination in people aged over 65 has previously been reported [20].

Leukopenia is highly prevalent in SOT recipients due to the use of immunosuppressive drugs and prophylaxis against opportunistic infections [21,22]. Furthermore, leukopenia following SARS-CoV-2 infection in kidney transplant recipients has been reported [23]. In kidney transplant recipients, an increased number of adverse outcomes, including infectious complications, have been reported after a mild degree of decreased leukocyte count [24], and a leukocyte count of <3 × 10^9^ cells/L has been associated with an increased risk of invasive aspergillosis [25]. Furthermore, in SOT recipients, a mild degree of decreased leukocyte count has been found to be associated with an increased hazard of blood stream infection [26] making a decreased leukocyte count a clinically relevant outcome when monitoring vaccine side effects. At present, there are no studies on a decreased leukocyte count as a potential side effect of SARS-CoV-2 mRNA vaccines in SOT recipients. As the beneficial effects of vaccination greatly outweigh the risk of serious side effects [15,27,28], and as vaccines against new variants emerge, vaccination continues to be strongly recommended for high-risk groups like SOT recipients [29,30]. Hence, investigations into the side effects and potential harmful effects of vaccinations are necessary to inform both future vaccination guidelines and transplant clinicians.

We aim to explore whether SARS-CoV-2 mRNA vaccination and SARS-CoV-2 infection are associated with risk of a decreased leukocyte count. Furthermore, we aim to describe the prevalence of decreased leukocyte counts and to describe leukocyte counts in SOT recipients before and after SARS-CoV-2 mRNA vaccination and SARS-CoV-2 infection.

## 2. Methods

### 2.1. Study Design

This prospective observational cohort study is based on a study of immune responses to SARS-CoV-2 vaccinations and infections in adult SOT recipients [31,32,33]. From January 2021 through April 2021, adult kidney, lung, and liver transplant recipients followed at Copenhagen University Hospital, Rigshospitalet, were invited to participate in the VACCIM study at the time of their first SARS-CoV-2 vaccination. From July 2021, adult kidney, lung, and liver transplant recipients were invited to participate regardless of vaccination status. All participants were followed until March 2023. In this study we included SOT recipients who lived in the Capital Region of Denmark or Region Zealand throughout the study period and were at least one year from transplantation at the start of follow-up (these participants were considered to be in a stable phase with regard to immunosuppressive maintenance therapy). SOT recipients who received SARS-CoV-2 vaccines other than monovalent or bivalent BNT162b2 vaccines as their first SARS-CoV-2 vaccine dose were excluded. In Denmark, SOT recipients were prioritized to receive BNT162b2 vaccines as their first SARS-CoV-2 vaccine. The first SARS-CoV-2 vaccines in Denmark were administered on 27 December 2020. To start observation at least two months prior to the administration of the first vaccine doses follow-up started on 27 October 2020. Follow-up ended in March 2023, after administration of SARS-CoV-2 vaccines other than monovalent or bivalent BNT162b2 vaccines, death, or re-transplantation, whichever came first. Data were available until March 2023, and participants were followed for the longest time possible to allow for the longest possible follow-up and the administration of as many vaccine doses and leukocyte samples as possible.

In Denmark, SOT recipients are monitored closely with routine biochemistry controls, including leukocyte counts, every 3–6 months. Leukocyte count is a part of the routine hematological panel. This study did not interfere with the Danish COVID-19 vaccination strategy or the sampling frequency of clinical samples from study participants. If a participant followed recommendations by the Danish health authorities during the study, the participant would receive five vaccine doses by the end of follow-up. Additionally, it was possible for clinicians to refer patients to receive extra vaccinations.

All participants provided written and oral informed consent. The study was conducted in accordance with the Declaration of Helsinki and approved by the Regional Scientific Ethics Committee of the Capital Region of Denmark (H-20079890).

### 2.2. Data Retrieval

Data on vaccinations were collected from the Danish Vaccination Registry (DDV). Since 2015, it has been mandatory to register all vaccines administered in Denmark in the DDV [34]. Data on leukocyte count were retrieved from the laboratory module of the electronic medical records system that covers all hospitals in the Capital Region of Denmark and Region Zealand. Data on demographics, immunosuppressive maintenance therapy, and transplantation-related clinical characteristics were collected from the electronic medical records system. Data on SARS-CoV-2 infections were collected from the Danish Microbiology Database (MiBa), which is a national database including information on all SARS-CoV-2 PCR samples from the primary sector, hospitals, and SARS-CoV-2 test centers in Denmark [35].

### 2.3. Definitions

We defined a decreased leukocyte count as a leukocyte count below <3.5 × 10^9^ cell/L following Common Terminology for Adverse Events (CTCAE) 5.0 [36]. According to CTCAE 5.0, a decreased leukocyte count can be classified as follows: grade 1 (lower limit normal to 3.0 × 10^9^ cell/L), grade 2 (<3.0–2.0 × 10^9^ cells/L), grade 3 (<2.0–1.0 × 10^9^ cell/L), and grade 4 (<1.0 × 10^9^ cells/L).

### 2.4. Self-Controlled Case Series Analysis

For the self-controlled case series (SCCS) analysis, we included all participants with a decreased leukocyte count during the study period. Risk periods were defined as days 1–21 after the first vaccine dose and days 1–60 after each SARS-CoV-2 mRNA vaccine dose. The shorter period after the first vaccine dose was due to the recommended interval of 21 days between the first and second vaccine doses. Control periods were defined as time outside of risk periods.

To investigate the relative incidence of decreased leukocyte counts after SARS-CoV-2 infection, we fitted a model with risk periods defined as days 0–60 after positive SARS-CoV-2 PCR tests.

### 2.5. Statistics

Continuous data were reported as medians with interquartile ranges (IQRs). Categorical data were reported as counts and percentages. Normality of data was assessed by quantile–quantile plots. To report the leukocyte counts before and after each vaccine dose, geometric mean concentrations with 95% confidence intervals (CIs) were calculated. To investigate whether there was a change in leukocyte count from before to after each vaccine dose, a linear mixed model with a compound symmetry covariance structure, random intercept, and slope was fitted using the lmm function from the LMMstar package [37]. The dependent variable of the model was the log-transformed leukocyte count; the independent variable was the time of sampling in relation to vaccine doses. The model was chosen to account for repeated measures and missing data, with the latter being handled by maximum likelihood estimation. Prevalence of decreased leukocyte count was calculated based on number of individuals with decreased leukocyte count in the population two months before and after SARS-CoV-2 infections and after each vaccine dose. Only participants with paired samples available from before and after a vaccine dose or SARS-CoV-2 infection were included in prevalence calculations for the respective vaccine dose or infection. The 95% CIs were calculated using the exact method, and differences in prevalence were tested using McNemar’s test. As the time between the first and second vaccine doses was 21 days in most cases, prevalence was not calculated for the period between the first and second vaccine doses. Instead, prevalences two months before first vaccine dose and two months after second vaccine dose were compared. To explore if the number of vaccine doses administered modified the effect of vaccination and SARS-CoV-2 infection on the prevalence of decreased leukocyte count, a generalized estimating equation (GEE) model with a compound symmetry covariance structure was fitted. The dependent variable was the decreased leukocyte count (binary), the independent variables were the time of sampling relative to vaccination (binary, pre-, or post-vaccine/infection), the number of vaccine doses (continuous), and an interaction term between the timing of sample and number of vaccine doses. To explore whether the relative incidence of the first episode of decreased leukocyte count was higher after the administration of SARS-CoV-2 mRNA vaccines, an extension of the SCCS model that allows for event-dependent exposure was fitted. To accommodate a possible violation of the SCCS model’s assumption that future exposures do not depend on a prior event, where future vaccination may be deferred due to the occurrence of a decreased leukocyte count, the extension of the model was applied. Furthermore, we fitted a standard SCCS model to explore whether the relative incidence of first episode of decreased leukocyte count was higher after SARS-CoV-2 infections than in control periods.

## 3. Results

### 3.1. Cohort Characteristics

Out of the 277 SOT recipients included in the VACCIM cohort, we excluded 9 who did not live in the Capital Region of Denmark or Region Zealand throughout the study period, 1 who did not receive BNT162b2 as the first SARS-CoV-2 vaccine, and 39 who were transplanted less than one year prior to the start of follow-up. The study cohort thus consisted of 228 adult kidney, lung, and liver transplant recipients. The median age at baseline was 56 years (IQR 46–64), and 59.6% of participants were male. Of the 228 SOT recipients, 107 (46.9%) were kidney transplant recipients, 87 (38.2%) were liver transplant recipients, and 34 (14.9%) were lung transplant recipients. For more information on comorbidities and immunosuppressive maintenance therapy at the beginning of the follow-up, see Table 1.

At the end of the study, 6 participants (2.6%) had received six doses of the SARS-CoV-2 mRNA vaccine, while 143 (62.7%) received five doses, 54 (23.7%) received four doses, 22 (9.6%) received three doses, and 3 participants (1.3%) received two doses.

During follow-up, five participants (2.2%) underwent re-transplantation, and seven participants (3.1%) died (Table 1).

### 3.2. Relative Incidence and Prevalence of Decreased Leukocyte Count Before and After SARS-CoV-2 mRNA Vaccination

During 484 person-years of follow-up, we observed 28 first cases of a decreased leukocyte count, corresponding to an incidence rate of 5.8 per 100 person-years of follow-up (95% CI: 3.8–8.4). For the SCCS analysis, all 28 cases were included (64 person-years of follow-up, 47 person-years of control time, and 17 person-years of risk time). The risk of a decreased leukocyte count was not increased after vaccination compared to the control periods (incidence rate ratio (IRR): 0.6; 95% CI: 0.2–2.1; *p* = 0.461).

Paired data from before the first vaccine and after the second vaccine dose were available for 136 participants. In the period two months before the first vaccine dose, the prevalence of decreased leukocyte counts was 2.9% (95% CI: 0.8–7.4), while it was 1.4% (95% CI: 0.2–5.2) after the second vaccine dose (*p* = 0.617). Around the third vaccine dose, paired data were available for 113 participants. In the period two months before the third vaccine dose, the prevalence of decreased leukocyte counts was 0.9% (95% CI: 0.02–4.8) compared with 1.8% (95% CI: 0.2–6.2) after the third dose (*p* = 0.683). Paired data were available for 112 participants around the fourth vaccine dose. Two months before the fourth dose, the prevalence of decreased leukocyte counts was 0.9% (95% CI: 0.02–4.9), while it was 1.8% (95% CI: 0.2–6.3) after the fourth dose (*p* >0.999). Finally, around the fifth dose, paired data were available for 71 participants, and the prevalence of decreased leukocyte counts was 2.8% (95% CI: 0.3–9.8) before the fifth vaccine dose compared with 1.4% (95% CI: 0.03–7.6) after the fifth dose (*p* > 0.999) (Figure 1). To explore trends in the effect of vaccination on the prevalence of decreased leukocyte counts with an increasing number of vaccine doses, a GEE model including an interaction term between the timing of samples in relation to vaccination and the number of vaccine doses was fitted. The *p*-value for the interaction term was 0.376, indicating that the effect of vaccination on the prevalence of decreased leukocyte counts does not change significantly with the number of vaccine doses.

Of the 28 first events of decreased leukocyte counts, 20 (71%) were grade 1, 7 (25%) were grade 2, 1 (4%) was grade 3, and none were grade 4. Among the 28 individuals with episodes of decreased leukocyte counts, the highest grades observed throughout the study period were grade 1 in 12 (43%) individuals, grade 2 in 13 (46%) individuals, grade 3 in 2 (7%) cases, and grade 4 in 1 (4%) case.

### 3.3. Relative Incidence and Prevalence of Decreased Leukocyte Counts Before and After SARS-CoV-2 Infection

The risk of a decreased leukocyte count was higher 60 days after a PCR-confirmed SARS-CoV2 infection compared to the control periods (IRR: 7.1, 95% CI: 2.8–18.1, *p* < 0.001).

Paired data from before and after SARS-CoV-2 infection were available for 85 participants. In the period two months before SARS-CoV-2 infections, the prevalence of decreased leukocyte counts was 2.4% (95% CI: 0.3–8.2), while it was 5.9% (95% CI: 1.9–13.2) in the period two months after SARS-CoV-2 infections (*p* = 0.074) (Figure 2). To explore trends in the effect of SARS-CoV-2 on the prevalence of decreased leukocyte counts with an increasing number of vaccine doses, a GEE model including an interaction term between the timing of samples in relation to SARS-CoV-2 infection and the number of vaccine doses was fitted. The *p*-value for the interaction term was 0.310, indicating that the effect of SARS-CoV-2 infection on the prevalence of decreased leukocyte counts does not change significantly with the number of vaccine doses.

### 3.4. Geometric Mean of Leukocyte Count Before and After Vaccination

Two months prior to the first vaccine dose, the geometric mean leukocyte count was 7.3 × 10^9^ cells/L (95% CI 6.9–7.6) (Table 2). From before to after the first vaccine dose, the geometric mean leukocyte count increased by 2.2% (95% CI: −2.1; +6.8, *p* = 0.319) (Table 3). Furthermore, the geometric mean leukocyte count decreased by 0.5% from before the first dose to after the second vaccine dose (95% CI: −3.9; +3.1, *p* = 0.799).

From two months before to two months after the third vaccine dose, the geometric mean leukocyte count increased by 2.9% (95% CI: −0.8; +6.7, *p* = 0.120), and from before to after the fourth vaccine dose, the geometric mean leukocyte count decreased by 1.8% (95% CI: −5.4; +1.9, *p* = 0.334). Finally, the geometric mean leukocyte count decreased by 0.8% from before to after the fifth vaccine dose (95% CI: −3.7; +5.4, *p* = 0.741) (Table 3).

For an overview of the geometric mean leukocyte count and missing data at each sampling time point, see Table 2.

## 4. Discussion

We found no evidence to support an association between SARS-CoV-2 mRNA vaccination and risk of decreased leukocyte count and no statistically significant differences in prevalence before and after each vaccine dose in SOT recipients. However, we found SARS-CoV-2 infection to be associated with increased risk of a decreased leukocyte count. Furthermore, we examined the leukocyte count before and after each dose of SARS-CoV-2 mRNA vaccine and found no statistically significant changes from before to after vaccination regardless of dose.

We found no statistically significant differences between the prevalence of decreased leukocyte counts before and after any doses of the SARS-CoV-2 mRNA vaccine. Similarly, the risk of a decreased leukocyte count was not higher after vaccination than during the control periods. To our knowledge, no previous studies have examined the prevalence, incidence, or risk of a decreased leukocyte count after SARS-CoV-2 mRNA vaccines in SOT recipients. Sing et al. found that among individuals with no history of prior hematological abnormalities who attended hospitals in Hong Kong, the second dose of BNT162b2 was associated with an IRR of 2.21 for a decreased leukocyte count [15]. Several factors may explain this discrepancy [38]. First, Sing et al. defined a decreased leukocyte count as a leukocyte count of less than 4.0 × 10^9^ cell/L, a higher cutoff than in our study, potentially leading to a higher incidence of decreased leukocyte counts in their study. Second, Sing et al. only analyzed the first and second doses of BNT162b2 and analyzed each dose separately, while we investigated up to the fifth dose of monovalent or bivalent BNT162b2 vaccines without differentiating between doses. Finally, Sing et al. excluded participants receiving immunosuppressive treatment, while all participants in our study were receiving maintenance immunosuppressive treatment. Therefore, both beneficial and potentially harmful immunological responses to vaccines are expected to differ between the populations.

Notably, we observed an increased risk of a decreased leukocyte count in SOT recipients in the period after infection with SARS-CoV-2 compared to the control periods (IRR 7.1). This corroborates previous reports of leukopenia in kidney transplant recipients upon hospitalization with SARS-CoV-2 infection [25], and reports of leukopenia in individuals from Wuhan and New York upon hospitalization with SARS-CoV-2 infections in the early period of the pandemic [39,40]. Most cases of decreased leukocyte counts in our data were grade 1 and 2 adverse events. Grade 1 adverse events, as defined by CTCAE, are mild and/or asymptomatic and do not require clinical intervention. While the clinical implications may be minor, a grade 1 decreased leukocyte count has previously been associated with adverse outcomes such as blood stream infections [24,26] and grade 2 has been associated with invasive aspergillosis [25] and should not be ignored. Several mechanisms, such as de novo development of autoantibodies against leukocytes, migration of leukocytes from the blood to peripheral tissues, viral infiltration and suppression of the bone marrow, and the use of antiviral drugs, may result in a decreased leukocyte count after SARS-CoV-2 infection. Although we cannot determine the precise mechanism behind the association between SARS-CoV-2 infection and leukocyte count based on data in the present study, the lack of association between SARS-CoV-2 vaccination and a decreased leukocyte count suggests that the explanation is not to be found in exposure to the SARS-CoV-2 spike antigen which mRNA in BNT162b2 vaccines encode.

We examined leukocyte counts before and after up to five doses of SARS-CoV-2 mRNA vaccine and found no statistically significant or clinically relevant changes in the leukocyte count from before to after vaccination. To our knowledge, there are no previous reports on the leukocyte count before and after SARS-CoV-2 mRNA vaccination in SOT recipients. In line with our results, a study including 36 individuals who received either one dose of BNT162b2 or one dose of AZD1222 SARS-CoV-2 vaccine found that BNT162b2 did not induce a change in leukocyte counts. In contrast, the adenovector-based AZD1222 vaccine induced a decrease in leukocyte counts [41].

The strengths of our study include leukocyte counts in a well-described and large cohort of SOT recipients, with data from DDV and MiBa, which are national databases providing detailed and complete information on SARS-CoV-2 vaccines and infections, respectively. Furthermore, we have data on the leukocyte count for up to five vaccine doses, providing information relevant in a real-world setting, where most SOT recipients have received more than two vaccine doses. Another strength is the use of SCCS analyses, a statistical method originally developed to investigate rare adverse events after vaccines.

This study also had some limitations. One limitation in this study was the lack of an unvaccinated SOT recipient control group to allow for a comparison of the development in leukocyte count over time in the absence of SARS-CoV-2 mRNA vaccines. Furthermore, the lack of a vaccinated healthy control group hinders any conclusion regarding whether the development in leukocyte count and the incidence of leukopenia after SARS-CoV-2 mRNA vaccines are comparable in SOT recipients and healthy individuals. Finally, this study was limited by missing data and a risk that our sampling may be unbalanced, as the leukocyte count samples were not pre-scheduled and more ill patients may be prone to having additional samples on clinical indication.

Future studies to understand the mechanism behind the impact of SARS-CoV-2 infections on the leukocyte count in SOT recipients and its clinical implications are warranted. Additionally, studies on the leukocyte count after SARS-CoV-2 vaccination in SOT recipients with pre-scheduled blood sampling before and after vaccination would be of value. Finally, it remains important to continue studying potential adverse effects to vaccines in all populations to provide the best foundation for decision making by policymakers, clinicians, and patients.

## 5. Conclusions

In conclusion, we did not find SARS-CoV-2 mRNA vaccination to be associated with risk of a decreased leukocyte count or to affect the prevalence of decreased leukocyte counts. In contrast, SARS-CoV-2 infection was associated with a higher risk of a decreased leukocyte count, and the prevalence of decreased leukocyte counts was higher after SARS-CoV-2 infection than before infection. Furthermore, we did not find SARS-CoV-2 mRNA vaccination to impact leukocyte counts.

## Figures and Tables

**Figure 1 vaccines-13-00103-f001:**
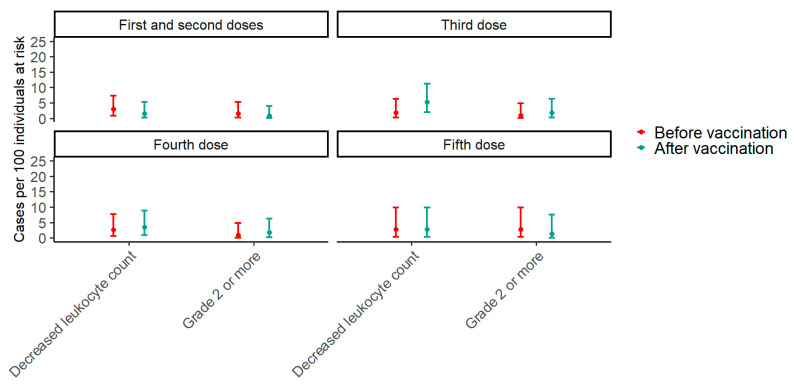
The prevalence of decreased leukocyte counts two months before and after each SARS-CoV-2 vaccine dose. The plot shows estimates of prevalence with 95% confidence intervals of any decreased leukocyte count and a grade 2 or higher decrease in leukocyte count two months before and after each SARS-CoV-2 mRNA vaccine dose. For the first and second doses, the “before vaccination” period is the period two months before the first dose, and the “after vaccination” period is the period two months after the second dose. Red represents before vaccination, and blue represents after vaccination. Around the first and second doses, paired data were available for 136 participants; around the third dose, paired data were available for 113 participants; around the fourth dose, paired data were available for 112 participants; and around the fifth dose, paired data were available for 71 participants.

**Figure 2 vaccines-13-00103-f002:**
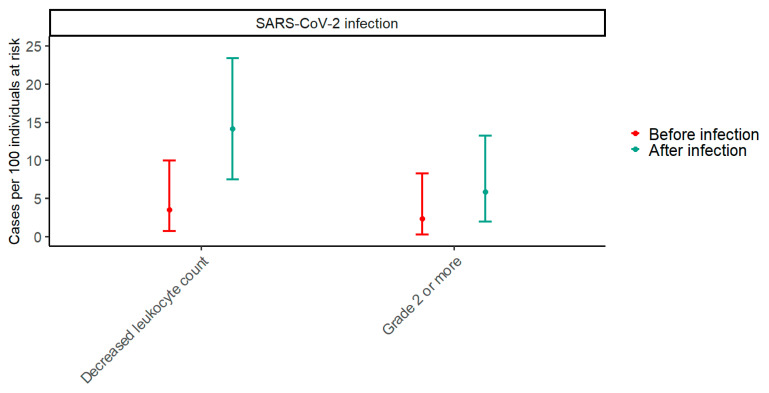
The prevalence of decreased leukocyte counts two months before and after SARS-CoV-2 infection. The plot shows estimates of prevalence with 95% confidence intervals of any decreased leukocyte count and a grade 2 or higher decrease in leukocyte count two months before and after SARS-CoV-2 infection. Red represents before infection, and blue represents after infection. Paired data with samples from both before and after SARS-CoV-2 infection were available for 85 participants.

**Table 1 vaccines-13-00103-t001:** The baseline characteristics of the study cohort.

Overall Study Population	*n* = 228
Age at baseline, median [IQR] in years	56 [46,64]
Male sex, *n* (%)	136 (59.6)
Time from transplantation to baseline, median [IQR] in years	7.4 [4,12]
Transplant type, *n* (%)	
Kidney	107 (46.9)
Liver	87 (38.2)
Lung	34 (14.9)
Re-transplantation, *n* (%)	25 (11.0)
Comorbidities, *n* (%)	
Diabetes mellitus	49 (21.5)
Cardiovascular disease	157 (68.9)
Dialysis	2 (0.9)
Immunosuppressive maintenance therapy, *n* (%)	
Tacrolimus	150 (65.8)
Ciclosporin	54 (23.7)
mTOR inhibitor	34 (14.9)
Corticosteroids	172 (75.4)
Antimetabolites	
None	34 (14.9)
Azathioprine	35 (15.4)
Mycophenolate	159 (69.7)
Number of vaccines at study end, *n* (%)	
2	3 (1.3)
3	22 (9.6)
4	54 (23.7)
5	143 (62.7)
6	6 (2.6)
Re-transplanted during follow-up, *n* (%)	5 (2.2)
Died during follow-up, *n* (%)	7 (3.1)

Abbreviations: IQR: interquartile range.

**Table 2 vaccines-13-00103-t002:** Geometric mean leukocyte counts and missing sample data before and after each vaccine dose.

Timing of Sample	Sample Count, *n*	Missing Samples out of Total Cohort, n (%)	Leukocyte Count (GMC)	95% CI
Pre-first vaccine dose	162	66 (28.9%)	7.3	6.9–7.6
Post-first vaccine dose	93	136 (59.2%)	7.5	7.0–7.9
Post-second vaccine dose	173	55 (24.1%)	6.9	6.9–7.6
Pre-third vaccine dose	169	59 (25.9%)	7.0	6.7–7.3
Post-third vaccine dose	158	70 (30.7%)	7.1	6.7–7.5
Pre-fourth vaccine dose	146	82 (36.0%)	7.2	6.8–7.5
Post-fourth vaccine dose	155	73 (32.0%)	7.2	6.8–7.6
Pre-fifth vaccine dose	104	124 (54.4%)	6.9	6.6–7.4
Post-fifth vaccine dose	103	125 (54.8%)	7.2	6.8–7.7
As the pre-second dose time intervals overlap with the time interval for post-first dose, no samples were categorized as “pre-second dose”.				

Abbreviations: GMC: geometric mean concentration; CI: confidence interval.

**Table 3 vaccines-13-00103-t003:** Percentage of change in white blood cell count between each vaccine dose.

Time Points	Change in Leukocyte Count (%)	95% CI	*p*-Value
Pre-first vs. post-first vaccine dose	2.2%	(−2.1; +6.5)	0.319
Pre-first vs. post-second vaccine dose	0.5%	(−3.9; +3.1)	0.799
Pre-third vs. post-third vaccine dose	2.9%	(−0.8; +6.7)	0.120
Pre-fourth vs. post-fourth vaccine dose	−1.8%	(−5.4; +1.9)	0.334
Pre-fifth vs. post-fifth vaccine dose	−0.8%	(−3.7; +5.4)	0.741

Abbreviations: WBC: white blood cell count; GMC: geometric mean concentration; CI: confidence interval.

## Data Availability

The data are not publicly available due to privacy or ethical restrictions. The data that support the findings of this study are available from the corresponding author upon reasonable request.

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
