# Peer review of "Leukocyte Count in Solid Organ Transplant Recipients After SARS-CoV-2 mRNA Vaccination and Infection"

_vaccines, 2025, doi:10.3390/vaccines13020103_

Round 1

Reviewer 1 Report

Comments and Suggestions for Authors

This interesting study by Kern et al. examined changes in leukocyte counts following primary COVID-19 vaccination, as well as after SARS-CoV-2 infection, in solid organ transplant recipients. The study is both relevant and well-executed in its objectives and methodology.

A few minor comments:

1) I would specify in the introduction that the safety of the BNT162b2 vaccine has also been studied in immunosuppressed individuals who are not transplant recipients but, for example, those undergoing chronic immunosuppressive therapy.

2) In the statistical analysis section, did you conduct a normality test prior to deciding between the mean and median to report continuous variable data? If so, it may be helpful to specify this.

3) In the discussion, it might be beneficial to elaborate further on the pathophysiological mechanism underlying the differences in leukopenic variations between vaccinated individuals and those directly infected with the virus.

Author Response

This interesting study by Kern et al. examined changes in leukocyte counts following primary COVID-19 vaccination, as well as after SARS-CoV-2 infection, in solid organ transplant recipients. The study is both relevant and well-executed in its objectives and methodology.

Thank you for taking the time to review the manuscript. We appreciate your comments and have tried to accommodate the manuscript to your suggestions to the best of our abilities. We hope you find the revised manuscript suitable for publication.

A few minor comments:

1) I would specify in the introduction that the safety of the BNT162b2 vaccine has also been studied in immunosuppressed individuals who are not transplant recipients but, for example, those undergoing chronic immunosuppressive therapy.

Thank you for this suggestion. We have added a sentence to the introduction about safety of BNT162b2 vaccines in patients undergoing chronic immunosuppressive therapy and included four new references.

Changes to the manuscript:

We added the following to the introduction, line 52:

“Furthermore, vaccination with BNT162b2 has also been reported to be safe in other immunocompromised populations than SOT recipients [11-14].

2) In the statistical analysis section, did you conduct a normality test prior to deciding between the mean and median to report continuous variable data? If so, it may be helpful to specify this.

Thank you for this excellent suggestion. We used quantile-quantile plots to assess normality of data. Information about this has been added to the statistics section.

Changes to the manuscript:

We added the following the statistics section, line 144:

“Normality of data was assessed by quantile-quantile plots”

3) In the discussion, it might be beneficial to elaborate further on the pathophysiological mechanism underlying the differences in leukopenic variations between vaccinated individuals and those directly infected with the virus

Thank you for this excellent suggestion. We have added the suggested details to the discussion. Furthermore, we added a suggestion on future research into a mechanism behind leukopenia after SARS-CoV-2 infection to the discussion.

Changes to the manuscript:

We added the following to the discussion section, line 276:

“Several mechanisms such as de novo development of autoantibodies against leukocytes, migration of leukocytes from blood to peripheral tissues, viral infiltration and suppression of the bone marrow, and use of antiviral drugs may result in decreased leukocyte count after SARS-CoV-2 infection. Although we cannot determine the precise mechanism behind the association between SARS-CoV-2 infection and leukocyte count based on data in the present study, the lack of association between SARS-CoV-2 vaccination and decreased leukocyte count suggests that the explanation is not to be found in exposure to the SARS-CoV-2 spike antigen which mRNA in BNT162b2 vaccines encode.”

And the following to the discussion section, line 309:

“Future studies to understand the mechanism behind the impact of SARS-CoV-2 infections on leukocyte count in SOT recipients and its clinical implications are warranted.”

Reviewer 2 Report

Comments and Suggestions for Authors

The authors definition of relative leucopenia is not acceptable and it is unclear what it is relative to. It would have been necessary to use neutropenia following internationally recognised reference ranges instead as it is neutropenia which predisposes to bacterial and fungal infections. COVID 19 causes lymphopenia and there is no evidence that this is clinically relevant. The lack of impact of the vaccine on lymphocyte or neutrophil numbers is already known.

Author Response

The authors definition of relative leucopenia is not acceptable and it is unclear what it is relative to. It would have been necessary to use neutropenia following internationally recognised reference ranges instead as it is neutropenia which predisposes to bacterial and fungal infections. COVID 19 causes lymphopenia and there is no evidence that this is clinically relevant. The lack of impact of the vaccine on lymphocyte or neutrophil numbers is already known.

Thank you for taking the time to review the manuscript. We appreciate your comments, [1] and we agree that the term relative leucopenia is inappropriate. In the Common Terminology Criteria for Adverse Events 5.0 (CTCAE) a grade 1 decrease in leukocytes is defined as leukocytes from below lower limit normal to 3.0  cells/L. Thus, in the revised manuscript, we have altered the terminology to comply with CTCAE and use the term decreased leukocyte count instead of relative leukopenia. As a grade 1 decrease in leukocyte count according to CTCAE corresponds to our former definition of outcome “relative leukopenia” the change does not affect the results. Furthermore, we have added information about CTCAE and a reference to CTCAE to the definitions section, and we provide information on cases of grade 2, 3 and 4 decreases in leukocyte counts in the result section. The grade 1 adverse event as defined by CTCAE is mild and/or asymptomatic and does not require clinical intervention. We have added a discussion of possible clinical implications of our finding of increased risk of decreased leukocyte count after SARS-CoV-2 infection to the discussion.

Lastly, as the focus of the manuscript is on leukocyte counts and not on neutrophile counts, we have made changes to the introduction to reflect this.

  1. Common Terminology Criteria for Adverse Events (CTCAE) | Protocol Development | CTEP Available online: https://ctep.cancer.gov/protocoldevelopment/electronic_applications/ctc.htm (accessed on 29 October 2024).

Changes to the manuscript:

The title has been changes to: “Leukocyte count in solid organ transplant recipients after SARS-CoV-2 mRNA vaccination and infection”

Throughout the manuscript “relative leukopenia” has been replaced with “decreased leukocyte count”

Changes to the introduction

In the introduction line 59-70 have been revised and now read:

“Furthermore, a decrease in leukocyte count after influenza vaccination in people aged over 65 has previously been reported [21].

Leukopenia is highly prevalent in SOT recipients due to use of immunosuppressive drugs and prophylaxis against opportunistic infections [22,23]. Furthermore, leukopenia after SARS-CoV-2 infections in kidney transplant recipients has been reported [24]. In kidney transplant recipients an increased number of adverse outcomes, including infectious complications, have been reported after a mild degree of decreased leukocyte count [25] and a leukocyte count of < 3  cells/L has been associated with an increased risk of invasive aspergillosis [26]. Furthermore, in SOT recipients a mild degree of decreased leukocyte count has been found to be associated with an increased hazard of blood stream infection [27] making decreased leukocyte count a clinically relevant outcome when monitoring vaccine side effects. At present, there are no published studies on decrease in leukocyte count as a potential side effect of SARS-CoV-2 mRNA vaccines in SOT recipients.”

Changes to the methods section

We have changed the definitions section:

“Definitions

We defined a decreased leukocyte count as a leukocyte count below <  cell/L following Common Terminology for Adverse Events (CTCAE) 5.0 [37]. According to CTCAE 5.0 a decreased leukocyte count can be classified as: grade 1 (lower limit normal to 3.0  cell/L), grade 2 (<3.0-2.0  cells/L), grade 3 (<2.0-1.0  cell/L), and grade 4 (<1.0  cells/L).

Changes to the results section

In the results we have elaborated with information on the grade of decreased leukocyte count cases and line 205-209 now reads:

“Of the 28 first events of decreased leukocyte count 20 (71%) were grade 1, seven (25%) were grade 2, one (4%) were grade 3 and none were grade 4. Among the 28 individuals with episodes of decreased leukocyte counts, the highest grade observed throughout the study period was grade 1 in 12 (43%) individuals, grade 2 in 13 (46%) individuals, grade 3 in two (7%) cases and grade 4 in one (4%) case.

Changes to the discussion

We added the following to the discussion, line 271-276:

“Most cases of decreased leukocyte count in our data were grade 1 and 2 adverse events. Grade 1 adverse event as defined by CTCAE are mild and/or asymptomatic and do not require clinical intervention. While the clinical implications may be minor, grade 1 decreased leukocyte count has previously been associated with adverse outcomes such as blood stream infection [25,27] and grade two has been associated with invasive aspergillosis [26] and should not be ignored.”

Reviewer 3 Report

Comments and Suggestions for Authors

Dear Authors,

read with interest your manuscript entitled "Leukopenia and leukocyte count in solid organ transplant recipients after SARS-CoV-2 mRNA vaccination and infection″. Your research conveys important findings about the effect of SARS CoV-2 mRNA vaccination and infection on leukocyte counts and the risk of relative leukopenia among SOT recipients. The use of a large cohort and data from national databases lends further credibility to your results. This investigation into leukocyte counts before and after vaccination, including the impact of SARS-CoV-2 infection, adds substantial value to the literature on the immunologic response to SARS-CoV-2 in this particular vulnerable population. The fact that SARS-CoV-2 mRNA vaccination did not increase the risk of relative leukopenia and did not lead to significant changes in leukocyte counts before and after vaccination was an important finding. On the other hand, the risk and prevalence of leukopenia are remarkably increased after SARS-CoV-2 infection, which corroborates earlier reports.
While the manuscript is overall well-executed and makes significant contributions to the field, I would like to propose some recommendations for further improvement.
You mentioned that SOT recipients in Denmark are monitored closely with routine biochemistry controls, including leukocyte counts every 3-6 months. It would be helpful to clarify that leukocyte counts are part of the routine hematological panel.
Your study, while it identifies the increased risk of leukopenia post-SARS-CoV-2 infection in SOT recipients, does not cover explanations in its underlying mechanism. It would be of value to discuss possible explanations for this mechanism.
You can suggest further directions of research with findings and limitations present.
In short, your study is a great contribution to the understanding of the immunological effects of SARS-CoV-2 mRNA vaccination and infection in SOT recipients, with great importance for the clinical implications. 

thank you for giving me the opportunity to review your work. I look forward to seeing the revised manuscript.

Author Response

Reviewer 3:

Dear Authors,
I read with interest your manuscript entitled "Leukopenia and leukocyte count in solid organ transplant recipients after SARS-CoV-2 mRNA vaccination and infection″. Your research conveys important findings about the effect of SARS CoV-2 mRNA vaccination and infection on leukocyte counts and the risk of relative leukopenia among SOT recipients. The use of a large cohort and data from national databases lends further credibility to your results. This investigation into leukocyte counts before and after vaccination, including the impact of SARS-CoV-2 infection, adds substantial value to the literature on the immunologic response to SARS-CoV-2 in this particular vulnerable population. The fact that SARS-CoV-2 mRNA vaccination did not increase the risk of relative leukopenia and did not lead to significant changes in leukocyte counts before and after vaccination was an important finding. On the other hand, the risk and prevalence of leukopenia are remarkably increased after SARS-CoV-2 infection, which corroborates earlier reports.
While the manuscript is overall well-executed and makes significant contributions to the field, I would like to propose some recommendations for further improvement.

Thank you for taking the time to review the manuscript. We appreciate your comments and have tried to accommodate the manuscript to your suggestions to the best of our abilities. We hope you find the revised manuscript suitable for publication.

1) You mentioned that SOT recipients in Denmark are monitored closely with routine biochemistry controls, including leukocyte counts every 3-6 months. It would be helpful to clarify that leukocyte counts are part of the routine hematological panel.

Response to reviewer 3 comment 1:

Thank you for pointing this out. We have now added to the manuscript, that leukocyte counts are a part of the routine hematological panel.

Changes to the manuscript:

We made the following addition to the method section, line 106-107:

"Leukocyte counts are a part of the routine hematological panel”

2) Your study, while it identifies the increased risk of leukopenia post-SARS-CoV-2 infection in SOT recipients, does not cover explanations in its underlying mechanism. It would be of value to discuss possible explanations for this mechanism.

Response to reviewer 3 comment 2:

Thank you for this excellent suggestion. Please see response to reviewer 1 question 3.

3) You can suggest further directions of research with findings and limitations present.

We agree and have added suggestions for future research.

Changes to the manuscript:

We added the following to the discussion line 309-315:

“Future studies to understand the mechanism behind the impact of SARS-CoV-2 infections on leukocyte count in SOT recipients and its clinical implications are warranted. Additionally, studies on leukocyte count after SARS-CoV-2 vaccination in SOT recipients with pre-scheduled blood-sampling before and after vaccination would be of value. Finally, it remains important to continue studying potential adverse effects to vaccines in all populations to provide the best foundation for decision making by policymakers, clinicians, and patients.”

In short, your study is a great contribution to the understanding of the immunological effects of SARS-CoV-2 mRNA vaccination and infection in SOT recipients, with great importance for the clinical implications. 

I thank you for giving me the opportunity to review your work. I look forward to seeing the revised manuscript.

Thank you for taking the time to review the manuscript. We appreciate your comments.

Reviewer 4 Report

Comments and Suggestions for Authors

I have reviewed the paper by Kern et al.

This is comparing data from 2021 vs 2023. There is no explanation on why these far apart times point were chosen.

There is no explanation why some patients received six or five doses of the mRNA vaccines.

Figure 1 x axis says time, which is not accurate. They just point out number of doses. There is no detail of time, even when (time in months) the leukocyte determination was done. Data should have been normalized according to time.

Author Response

I have reviewed the paper by Kern et al.

1) This is comparing data from 2021 vs 2023. There is no explanation on why these far apart times point were chosen.

Response to reviewer 4 comment 1:

Thank you for this important question, we agree that choice of study period could be clearer. The follow-up starts 2 months prior to the first SARS-CoV-2 vaccine being administered in Denmark. This was done to allow us to collect information about the leukocyte counts before the first vaccine dose. End of follow-up was chosen as the latest possible date to allow for as long follow-up and as many vaccine doses as possible.

Information has been added to the manuscript on the motivation behind the long follow-up period.

Changes to the manuscript:

The method section, 98-99 which previously read:

“To start observation prior to administration of the first vaccine doses participants were followed from October 27, 2020, …”

Now reads:

“To start observation at least two months prior to administration of the first vaccine doses follow-up started October 27, 2020. Follow-up ended….”

In the methods section, line 101-104 we have added:

“Data was available until March 2023 and participants were followed for the longest time possible to allow for as long follow-up and administration of as many vaccine doses and leukocyte samples as possible.”

2) There is no explanation why some patients received six or five doses of the mRNA vaccines.

Response to reviewer 4 comment 2:

Thank you for bringing this very important point to our attention. The study was purely observational and had no impact on vaccinations. SOT recipients were recommended booster doses by the Danish health authorities during the study period. The recommended vaccination program for SOT recipients resulted in 5 doses during the study period. However, some SOT recipients did not adhere to the program. Furthermore, clinicians were able to refer patients for an additional dose, if relevant. Hence the small number of participants with six vaccine doses.

Information about recommendations from the health authorities on doses and the possibility for clinicians to refer patients to vaccination has been added to the manuscript.

Changes to the manuscript:

We added the following to the methods section, line 108-111:

“If a participant followed recommendations by the Danish health authorities during the study the participant would receive five vaccine doses by end of follow-up. Additionally, it was possible for clinicians to refer patients to extra vaccinations.”

3) Figure 1 x axis says time, which is not accurate. They just point out number of doses. There is no detail of time, even when (time in months) the leukocyte determination was done. Data should have been normalized according to time.

Response to reviewer 4 comment 3:

Thank you for pointing this out. We have now corrected the x-axis label to state vaccine doses instead of time.

Changes to the manuscript:

Figure 1 now has a new x-axis label.

During revision of the manuscript, we also corrected a few typos and an error in description of the statistics:

Changes to statistics:

The statistics section line 157 which previously read:

“….and differences in prevalence were tested for using Fisher’s exact test.”

Now reads:

““….and differences in prevalence were tested for using McNemar’s test.”

Changes to the results:

In line 186 we have specified that we analyze first cases of decreased leukocyte count

In the results section line 192-201 p-values have been corrected, this was only minor changes which, did not change interpretation of the results:

And

The results section, line 214-216, which previously read:

“In the period two months before SARS-CoV-2 infections, the prevalence of relative leukopenia was 1.5% (95% CI: 0.2-5.2) while it was 8.0% (95% CI: 4.1-13.9) in the period two months after SARS-CoV-2 infections (p = 0.019).”

Now reads:

“In the period two months before SARS-CoV-2 infections, the prevalence of decreased leukocyte count was 2.2% (95% CI: 0.5-6.3) while it was 8.8% (95% CI: 4.6-14.8) in the period two months after SARS-CoV-2 infections (p = 0.008).”

Round 2

Reviewer 1 Report

Comments and Suggestions for Authors

The authors revised the manuscript.

Author Response

Thank you for taking the time to review the manuscript. Your comments and suggestions helped improve the quality of the work.

Reviewer 2 Report

Comments and Suggestions for Authors

The authors have not sufficiently improved the manuscript which would have required to focus only on neutrophil count.

Author Response

Reviewer 2 comment 1

The authors have not sufficiently improved the manuscript which would have required to focus only on neutrophil count.

Thank you again for taking the time to review the manuscript. We respectfully disagree. The aim of our manuscript is to explore if SARS-CoV-2 mRNA vaccination and SARS-CoV-2 infection are associated with increased risk of decreased leukocyte counts. Decreased leukocyte count is a relevant clinical outcome in solid organ transplant recipients, and even mild grades of decreased leukocyte count have been associated with increased risk of adverse outcomes such as infectious complications[1,2]. Furthermore, we explore decreased leukocyte counts as a potential adverse effect to COVID-19 vaccines. When adverse events are monitored in phase I-III trials of vaccines, change in leukocyte count is monitored exemplified by the protocols for the phase I-II trials of the SARS-CoV-2 mRNA vaccine candidates mRNA-1237 [3] BNT162b1 and BNT162b2 [4] and phase III trial of BNT162b2. Lastly, decreased leukocyte count after BNT162b2 vaccination has previously been reported in people without prior hematological diseases attending hospitals in Hong Kong [5], and decreased leukocyte count has previously been reported in kidney transplant recipients following SARS-CoV-2 infection. Thus, having leukocyte counts as our main endpoint allows comparison to other similar studies.

As SARS-CoV-2 mRNA booster vaccinations continue to be recommended for SOT recipients we strongly believe that investigations into potential adverse event such as decreased leukocyte counts are of high value for patients, clinicians, and policymakers to provide the best foundation for informed decision making.

We acknowledge that adding neutrophil counts would have provided additional information. However, neutrophil counts are not a part of routine monitoring of SOT recipients at our site, nor at many other sites. Thus, we cannot add the requested neutrophil counts. However, as argued above, we find that leukocyte counts are both relevant and adequate as the sole outcome, and it is our opinion that reporting leukocyte counts after SARS-CoV-2 mRNA vaccination in SOT recipients is highly relevant.

References:

  1. Henningsen, M.; Jaenigen, B.; Zschiedrich, S.; Pisarski, P.; Walz, G.; Schneider, J. Risk Factors and Management of Leukopenia After Kidney Transplantation: A Single-Center Experience. Transplant Proc 2021, 53, 1589–1598, doi:10.1016/J.TRANSPROCEED.2021.04.011.
  2. Møller, D.L.; Sørensen, S.S.; Perch, M.; Gustafsson, F.; Rezahosseini, O.; Knudsen, A.D.; Scheike, T.; Knudsen, J.D.; Lundgren, J.; Rasmussen, A.; et al. Bacterial and Fungal Bloodstream Infections in Solid Organ Transplant Recipients: Results from a Danish Cohort with Nationwide Follow-Up. Clin Microbiol Infect 2022, 28, 391–397, doi:10.1016/J.CMI.2021.07.021.
  3. Jackson, L.A.; Anderson, E.J.; Rouphael, N.G.; Roberts, P.C.; Makhene, M.; Coler, R.N.; McCullough, M.P.; Chappell, J.D.; Denison, M.R.; Stevens, L.J.; et al. An MRNA Vaccine against SARS-CoV-2 — Preliminary Report. New England Journal of Medicine 2020, 383, 1920–1931, doi:10.1056/NEJMoa2022483.
  4. Walsh, E.E.; Frenck, R.W.; Falsey, A.R.; Kitchin, N.; Absalon, J.; Gurtman, A.; Lockhart, S.; Neuzil, K.; Mulligan, M.J.; Bailey, R.; et al. Safety and Immunogenicity of Two RNA-Based Covid-19 Vaccine Candidates. N Engl J Med 2020, 383, 2439–2450, doi:10.1056/NEJMOA2027906.
  5. Sing, C.W.; Tang, C.T.L.; Chui, C.S.L.; Fan, M.; Lai, F.T.T.; Li, X.; Wan, E.Y.F.; Wong, C.K.H.; Chan, E.W.Y.; Hung, I.F.N.; et al. COVID-19 Vaccines and Risks of Hematological Abnormalities: Nested Case–Control and Self-Controlled Case Series Study. Am J Hematol 2022, 97, 470–480, doi:10.1002/AJH.26478.

Reviewer 3 Report

Comments and Suggestions for Authors

Thank you for responding to my suggestions in the revised manuscript. I appreciate your effort and attention to detail in making the necessary corrections. Your work is much improved and I congratulate you on your thorough approach.

Author Response

(The authors gave the same response as above.)

Reviewer 4 Report

Comments and Suggestions for Authors

I do not think the analysis is done correctly. McNemar test requires the use of paired data pre- and post- an intervention. Are they doing a McNemar between the pre- post- first, second, third tec dose? The numbers in the table show that these subjects are not paired.

A better approach is to do a trend analysis to see the effect of multiple doses.

Figure 1, does not make much sense, as it is supposed to show subjects with leukopenia. It would have been more useful to show the level of leukopenia, (as Table 2 does) and analyzed accordingly (which Table 2 did not).

Author Response

Please see the attachtment

Round 3

Reviewer 2 Report

Comments and Suggestions for Authors

The authors have not followed my request.

Author Response

As discussed with the editorial office, we have nothing to add to the responses given to Reviewer 2 in the second round of revisions and to the cover letter submitted with the second round of revisions.

Reviewer 4 Report

Comments and Suggestions for Authors

Thank you for rectifying your analysis of the McNemar test.

Given that the time periods before vs. after and the infection lenght differ between subjects, it is indispensable to perform a trend analysis to see the effect of multiple doses, as I suggested before.

Author Response

Thank you for rectifying your analysis of the McNemar test.

Given that the time periods before vs. after and the infection lenght differ between subjects, it is indispensable to perform a trend analysis to see the effect of multiple doses, as I suggested before.

Thank you for this excellent suggestion. We agree that doing a trend analysis adds valuable information to the manuscript. To explore if there is a trend in the effect of vaccination on the prevalence of decreased leukocyte count with increasing number of vaccines doses, we fitted a generalized estimating equation (GEE) model with compound symmetry covariance structure. The dependent variable was decreased leukocyte count (yes/no) and the independent variables were time of sample in relation to vaccination (dichotomous: pre or post vaccination) and number of vaccine doses (continuous) and an interaction term between timing of sample and number of vaccine doses. The p-value for the interaction term was 0.376 indicating that the effect of vaccination on the prevalence of decreased leukocyte count does not change significantly with increasing number of vaccine doses. We also fitted a model where we adjusted for the time from vaccination to sample (in days, continuous) and found similar results.

Furthermore, we fitted a GEE model with compound symmetry covariance structure, decreased leukocyte count (yes/no) as dependent variable, and time of sample in relation to SARS-Cov-2 infection (dichotomous, pre or post infection), number of vaccine doses (continuous) and an interaction term between timing of sample and number of vaccine doses as independent variables. The p-value for the interaction term was 0.310 indicating that the effect of SARS-CoV-2 infection on the prevalence of decreased leukocyte count does not change significantly with increasing number of vaccine doses. We have added the results of the analysis to the results section and describe the methods in the statistics section in the revised manuscript.

Lastly, we have made changes to the methods section to clarify that participants were observed for two months for all calculations of prevalences.

Changes to the methods

In the statistics section, line 153 we deleted:

“….except for before second dose, as this interval overlapped with the post-first vaccine interval”

And in the statistics section, line 158 we added:

“As the time between first and second vaccine dose was 21 days in most cases prevalence was not calculated for the period between first and second vaccine dose. Instead, prevalences two months before first vaccine dose and two months after second vaccine dose were compared”

We added the following to the statistics section, line 161

“To explore if the number of vaccine doses administered modified the effect of vaccination and SARS-CoV-2 infection on the prevalence of decreased leukocyte count a generalized estimating equation (GEE) model with compound symmetry covariance structure was fitted. The dependent variable was decreased leukocyte count (binary), the independent variables were time of sampling relative to vaccination (binary, pre or post vaccine/infection), number of vaccine doses (continuous) and an interaction term between timing of sample and number of vaccine doses.”

Changes to the results:

We added the following to the results section, line 214:

“To explore trends in the effect of vaccination on the prevalence of decreased leukocyte count with increasing number of vaccine doses, a GEE model including an interaction term between timing of sample and number of vaccine doses was fitted. The p-value for the interaction term was 0.376, indicating that the effect of vaccination on the prevalence of decreased leukocyte count does not change significantly with the number of vaccine doses”

And the following to the results section, line 242:

“To explore trends in the effect of SARS-CoV-2 on the prevalence of decreased leukocyte count with increasing number of vaccine doses, a GEE model including an interaction term between timing of sample in relation to SARS-CoV-2 infection and number of vaccine doses was fitted. The p-value for the interaction term was 0.310, indicating that the effect of SARS-CoV-2 infection on the prevalence of decreased leukocyte count does not change significantly with the number of vaccine doses”